# FedPMVR: Addressing Data Heterogeneity in Federated Learning through Partial Momentum Variance Reduction

## Abstract

Federated learning (FL) emerges as a promising paradigm for training machine learning models on decentralized data sources while preserving privacy. However, the presence of not independent and identically distributed (non-IID) data among the clients introduces high variance in gradient updates, posing a significant challenge to the global model's performance in terms of accuracy and convergence. To mitigate the adverse effects of data heterogeneity, we propose a novel momentum-based partial variance reduction technique. Our approach adjusts the gradient updates for the final classification layers of the client's neural network by leveraging the gradient differences between local and global models. This adjustment aims to effectively capture and mitigate client drift, a key challenge arises from the presence of non-IID data distributions across clients. We systematically explains client drifts and conduct extensive experiments on three widely-used datasets, demonstrating that our method significantly enhances global model accuracy while reducing the communication rounds needed for convergence. Notably, our momentum-based partial variance reduction technique provides a robust mechanism, rendering more efficient and effective in scenarios with inherently non-IID and heterogeneous data distributions. By addressing the critical challenge of data heterogeneity in FL, our proposed approach paves the way for more reliable and accurate model training while preserving the privacy of decentralized data sources. The code is available at the following link https://anonymous.4open.science/r/FedPMVR-33C1.

## 1 Introduction

Federated learning (FL) has emerged as a effective solution to privacy concerns in centralized model training, enabling multiple clients to collaboratively build while keeping their raw data decentralized and private (McMahan et al., 2017), (Guo et al., 2021), (Park et al., 2021). In FL, clients perform local training on their data and only share model updates (weights) with a central server, which aggregates these updates to refine the global model. While federated training offers significant advantages in preserving user privacy, it faces a practical obstacle in the form of data heterogeneity (Li et al., 2020a). Diverse user behaviors across different clients can lead to significant heterogeneity in their local data, resulting in unstable convergence, slow training progress, and suboptimal or even detrimental model performance (Zhao et al., 2018), (Karimireddy et al., 2020). FedAvg (McMahan et al., 2017), the widely adopted FL algorithm, often encounters challenges in achieving optimal accuracy and convergence, particularly in scenarios with heterogeneous data distributions across clients. This difficulty arises from *client drift*, a phenomenon resulting from the varying nature of data among participating clients. Client drift occurs when local models diverge significantly from the global model, causing aggregated updates to become less effective or detrimental to overall performance (Karimireddy et al., 2020). Addressing data heterogeneity and client drift is crucial for harnessing the full potential of FL in real-world applications with decentralized and diverse data.

Previous research efforts have aimed to address the issue of client drift by introducing penalties for the divergence between client and server models (Li et al., 2020a), (Li et al., 2021a) or by employing variance reduction approaches during the client model update process (Karimireddy et al., 2020), (Acar et al., 2021). Techniques like CCVR (Luo et al., 2021) first reported that classification

layers are solely accountable for client drift and addressed data heterogeneity through re-training classifiers using virtual features derived from an approximate Gaussian mixture model (GMM). However, creating representative datasets or features spanning multiple clients can be challenging, requiring domain knowledge and raising privacy concerns. Recent research has highlighted how biased classifiers adversely affect the performance of federated training on heterogeneous and long-tailed data (Shang et al., 2022). To mitigate this issue, Luo et al. (2021) proposed a method that addresses long-tailed and heterogeneous data by re-training the classifier on the server using learn-able features from client models. Additionally, researchers have investigated gradient variability across clients, especially in deeper or classification layers, using metrics like drift diversity. In particular, Li et al. (2023) found that client drift predominantly originates from the classification layers and proposed a partial variance reduction technique using control variates, though this approach can lead to increased communication costs.

Building on prior researches (Refer to Appendix A for the detailed related works), we introduce Federated Partial Momentum Variance Reduction (FedPMVR) that integrates a momentum-based variance reduction technique that selectively targets the classification layers of client neural networks. Our approach employs standard stochastic gradient descent (SGD) in the initial layers to capture diverse representations from heterogeneous client data while incorporating a momentum term in the classification layers to enhance gradient alignment. By maintaining a local momentum term for each client, FedPMVR captures the drift between local and global models, efficiently addressing client drift in non-IID data settings. To the best of our knowledge, this is the first work to leverage such a selective momentum-based regularization, achieving improved performance by balancing representation learning with variance reduction. The key contributions are summarized as follows:

- We propose FedPMVR, which employs momentum terms to reduce divergence between the classification layers of client and global models, effectively addressing client drift from non-IID data distributions.

- Momentum term incorporation in FedPMVR allows aligning local models with the global model, facilitating faster convergence to the global optimum and accelerating convergence.

- We provide theoretical convergence guarantees for FedPMVR in convex and non-convex settings, demonstrating its limited reliance on measures of data heterogeneity.

- Experimental evaluations consistently show that FedPMVR outperforms state-of-the-art approaches across diverse datasets with varying levels of data heterogeneity.

## 2 METHODS AND MATERIALS

This section explores the impact of client drift resulting from data heterogeneity, introduces the proposed FedPMVR approach, highlights the role and benefits of momentum terms, and discusses how partial momentum variance helps mitigate client drift, alongside a theoretical convergence analysis.

### 2.1 ILLUSTRATION OF CLIENT DRIFT

Considering a standard FL setup where the global model at round $t$ is denoted by $W_t$. Each client $i$ performs local updates using gradient descent with the following update rule:

$$w_t^i = w_{t-1}^i - \eta \nabla L_i(w_{t-1}^i), \tag{1}$$

where $w_t^i$ is the local model of client $i$ at iteration $t$, $\eta$ is the local learning rate, and $\nabla L_i(w_{t-1}^i)$ is the gradient of the loss function for client $i$ at iteration $t-1$. The deviation of each client model from the global optimum $W^*$ is captured by the distance $\zeta_i = \|w_R^i - W^*\|$, where $R$ corresponds the number of local updates. With increased heterogeneity (lower $\bar{\beta}$), the distance $\zeta_i$ grows, leading to greater client drift. The server-side aggregation of client models is given by:

$$W_{t+1} = \frac{1}{C} \sum_{i=1}^{C} w_t^i, \tag{2}$$

where $C$ is the number of clients. Increased drift caused by data heterogeneity adversely affects the outcomes of model aggregation, ultimately hindering the convergence of the global model to the optimal solution $W^*$. For example, consider a scenario with two clients optimizing simple quadratic loss functions, both aiming for a global optimum of $W^* = 3$. If we set the initial global model as $W^0 = 0$, the disparity in data distributions can significantly slow down the rate at which the global model approaches this optimal point, illustrating the impact of client drift on model training dynamics.

**Case 1: Low Heterogeneity** In this scenario, the clients' data distributions are more homogeneous, meaning that the data across clients is relatively similar. Consider Client 1 with a loss function $L_1(w) = (w - 15)^2$, resulting in a local gradient $\nabla L_1(w) = 2(w - 15)$. The corresponding weight update is $w_t^1 = w_{t-1} - \eta \cdot 2(w_{t-1} - 15)$. Similarly, for Client 2, with a loss function $L_2(w) = (w - 2)^2$, the gradient becomes $\nabla L_2(w) = 2(w - 2)$, and the update rule follows as $w_t^2 = w_{t-1} - \eta \cdot 2(w_{t-1} - 2)$. Starting with $W^0 = 0$, after one round of updates, the client models are updated as follows:

- Client 1: $w_1^1 = 0 - \eta \cdot 2(0 - 15) = 30\eta$,
- Client 2: $w_2^1 = 0 - \eta \cdot 2(0 - 2) = 4\eta$.

Here the loss functions for the clients were deliberately chosen with significantly different optima (15 and 2) to allow each client to independently explore their local minima. This setup facilitates convergence towards the global optimum during federated training, enabling better alignment with the overall objective. At the server, the aggregation of client models can be expressed as follows:

$$w^1 = \frac{1}{2}(30\eta + 4\eta) = 17\eta. \tag{3}$$

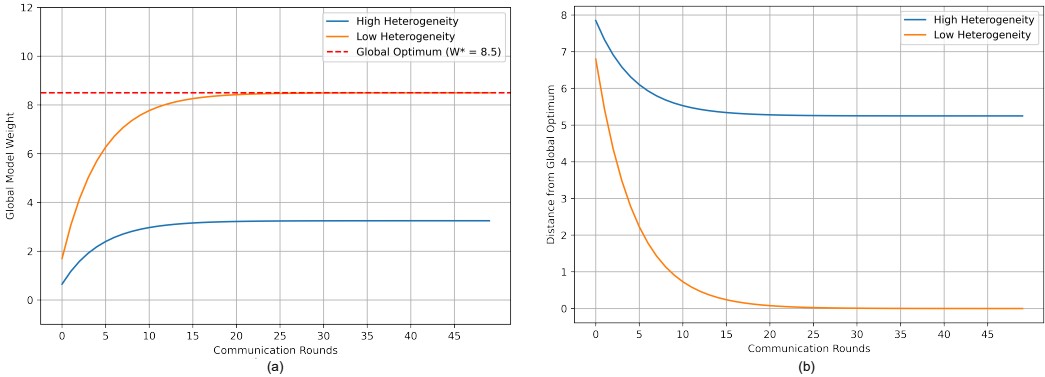

(a)                    (b)

Figure 1: Illustration of client drift under data heterogeneity: (a) shows the global model weights across communication rounds compared to the optimum global model weight, and (b) depicts the distance between the obtained global model and the true global optimum after each round.

**Case 2: High Heterogeneity** In this scenario, clients exhibit highly diverse data distributions, indicating significant variations in their local datasets. Consider Client 1 with loss function $L_1(w) = (w - 3)^2$, with the local gradient $\nabla L_1(w) = 2(w - 3)$. Similarly, for Client 2, with a loss function $L_2(w) = (w - 3.5)^2$, the local gradient $\nabla L_2(w) = 2(w - 3.5)$. Starting with $w^0 = 0$, after one round of updates, the client models are updated as follows:

- Client 1: $w_1^1 = 0 - \eta \cdot 2(0 - 3) = 6\eta$,
- Client 2: $w_2^1 = 0 - \eta \cdot 2(0 - 3.5) = 3.5\eta$.

The loss functions for the clients were taken with closely aligned optima (3 and 3.5), enabling them to converge quickly to their local minima during federated training. However, this rapid convergence is likely to occur far from the true global optimum. At the server, the models are aggregated and expressed as follows:

$$w^1 = \frac{1}{2}(6\eta + 3.5\eta) = 4.75\eta. \tag{4}$$

The above analysis can be similarly applied to subsequent training rounds. The outcomes illustrated in Fig.1 (a) indicates that with increased data heterogeneity, the model rapidly converges to a stable point rather than achieving the global optimum. Conversely, in scenarios of lower heterogeneity, convergence takes additional rounds, ultimately leading to the global optimum. A similar trend is evident in Fig.1 (b).

## 2.2 FedPMVR: Federated Partial Momentum Variance Reduction

Our proposed method, FedPMVR, incorporates momentum correction for the last classification layers of the neural network during the local model updation. The core concept is to harness the benefits of momentum-based optimization on the local client side to mitigate the effects of non-IID data distribution across clients, which can lead to better performance. Details of the two-step process are given below.

### 2.2.1 Client Update Step

In the client update step, each client performs the following operations:

- **Local Model Initialization:** To ensure a consistent starting point for local training, each client initializes its local model by setting the weight parameters to those of the global model received from the central server. This initialization process guarantees that all clients start their local optimization from an identical model state.

- **Local Model Training:** For a predefined number of epochs, the client trains the local model using their local dataset. This training process involves updating the model weights using SGD or a variant of SGD, such as Adam or RMSprop.

- **Gradient Computation:** Upon completing local model training, the client calculates the gradients of the updated local weights in relation to the initial global weights. These gradients indicate the necessary weight adjustments to minimize the loss function for the client's local data.

- **Momentum Update:** For the last few layers of the neural network, the client updates the corresponding momentum terms using an exponential moving average of the gradients. Specifically, for each weight in these layers, the momentum term is updated as follows:

$$m^{'} = \alpha \cdot \text{gradient} + (1 - \alpha) \cdot m, \tag{5}$$

  where $m^{'}$ is the momentum term, $\alpha$ is a hyperparameter that controls the learning rate for the momentum update, 'gradient' denotes the current weight gradient, and $m$ is the previous momentum term value.

- **Weight Correction:** After updating the momentum terms, the client corrects the local model weights for the last few layers by subtracting the corresponding momentum terms from the weights. This weight correction step is performed as follows:

$$w'' = w' - m', \tag{6}$$

  where $w''$ is the corrected weights and $w'$ is the previous weights.

### 2.2.2 Server Aggregation

On the server side, the local model weights from the all clients are averaged to obtain a new global model using Eq. 7, where $\mathbf{W_{t+1}}$ is the global model for round $t + 1$, $C$ is the number of clients, $n_c$ is the number of samples for client $c$, $n$ is the total number of samples across all clients. and $w_t^i$ is the $i^{th}$ client weighs at round $t$.

$$\mathbf{W}_{t+1} = \frac{n_c}{n} \sum_{i=1}^{C} \mathbf{w}_t^i \tag{7}$$

The server then updates the global model weights with the computed averages, and the aggregation process is repeated for the desired number of communication rounds. The process is outlined in Algorithm 1 in the Appendix.

## 2.3 Usefulness of Momentum term for Each Client

Incorporating momentum terms into the FL framework, particularly at the client level, presents a robust solution to mitigate the issue of client drift due to data heterogeneity. It stabilizes gradient updates by accumulating past gradient information, which counters sudden changes and reduces variance caused by non-IID data across clients. This stabilization leads to more reliable updates and mitigates client drift. Additionally, momentum accelerates convergence by maintaining the gradient direction and reducing oscillations, allowing clients to overcome local minima and saddle points more effectively. An example of integrating the proposed approach into the standard FedAvg algorithm is provided below, demonstrating how partial momentum variance reduction effectively mitigates client drift caused by heterogeneous data distributions.

In FedAvg, each client $c$ performs local updates based on its local data distribution, which may differ significantly from the other client's data distribution. The objective is to minimize the global loss function as presented in Eq. 8:

$$\min_{\mathbf{w}} f(\mathbf{w}) = \sum_{c=1}^{C} \frac{n_c}{n} f_c(\mathbf{w}), \tag{8}$$

where $f_c(\mathbf{w})$ is the local loss function for client $c$, $n_c$ is the number of samples for client $c$, and $n$ is the total number of samples across all clients. Each client performs several gradient descent steps on its local loss function. For client $c$, at communication round $t$, the local update can be presented as Eq. 1. After the local updates, the server aggregates the local models by averaging the local updates using Eq. 7. The global update can be written as below:

$$\mathbf{w}_{t+1} = \mathbf{w}_t - \eta_g \sum_{c=1}^{C} \frac{n_c}{n} \nabla f_c(\mathbf{w}_t). \tag{9}$$

where $\eta_g$ is the global learning rate. This equation shows that the global update depends on a weighted sum of local gradients. When the data distribution of client $c$ deviates significantly from that of other clients, the local gradient $\nabla f_c(\mathbf{w}_t)$ will differ from the true global gradient $\nabla f(\mathbf{w}_t)$. This discrepancy, termed as client drift $\delta_c$ is defined in Eq. 10:

$$\delta_c = \nabla f_c(\mathbf{w}_t) - \nabla f(\mathbf{w}_t). \tag{10}$$

Thus, the aggregated global update is expressed as:

$$\mathbf{w}_{t+1} = \mathbf{w}_t - \eta \left( \nabla f(\mathbf{w}_t) + \sum_{c=1}^{C} \frac{n_c}{n} \delta_c \right). \tag{11}$$

The term $\sum_{c=1}^{C} \frac{n_c}{n} \delta_c$ represents the *drift* caused by non-IID data across clients. When local gradients are highly variable due to data heterogeneity, this drift term increases, causing divergence between the global model and the optimal solution.

### 2.3.1 Mitigating Client Drift Using Partial Momentum Variance Reduction

In the proposed method, momentum is applied exclusively to the last two layers of each client's local model. Denoting the momentum term for the $i^{th}$ layer of client $c$ at round $t$ as $m_t^{c,i}$, the momentum-based local update for the last two layers ($i = -2, -1$) is defined as:

$$m_{t+1}^{c,i} = \alpha m_t^{c,i} + (1 - \alpha)\nabla f_c(\mathbf{w}_t^{c,i}), \tag{12}$$

where $\alpha \in [0, 1)$ is the learning rate for momentum term, and $i$ denotes the layer index. Consequently, the weight update for the last two layers is expressed as:

$$\mathbf{w}_{t+1}^{c,i} = \mathbf{w}_t^{c,i} - \eta m_{t+1}^{c,i}, \quad \text{for } i = -2, -1. \tag{13}$$

For all other layers ($i \neq -2, -1$), the model updates follow the standard SGD process:

$$\mathbf{w}_{t+1}^{c,i} = \mathbf{w}_t^{c,i} - \eta \nabla f_k(\mathbf{w}_t^{c,i}), \quad \text{for } i \neq -2, -1. \tag{14}$$

The global update integrates momentum for the final two layers while employing standard SGD for the others. Specifically, the global update for the $i^{th}$ layer is expressed as follows:

$$\mathbf{w}_{t+1}^i = \begin{cases} \mathbf{w}_t^i - \eta \sum_{c=1}^{C} \frac{n_c}{n} m_{t+1}^{c,i}, & \text{if } i = -2, -1, \\ \mathbf{w}_t^i - \eta \sum_{c=1}^{C} \frac{n_c}{n} \nabla f_c(\mathbf{w}_t^i), & \text{otherwise.} \end{cases} \tag{15}$$

Incorporating momentum exclusively in the last two layers significantly reduces variability in local updates by smoothing out gradient fluctuations, particularly in layers most affected by non-IID data. This smoothing effect minimizes drift in these deeper layers, which are crucial for overall performance, while simultaneously aligning the updates with the global objective, thereby reducing client drift and improving model consistency. Mathematically, the drift for the last two layers is modified to:

$$\delta_c^i = (1 - \alpha) \left( \nabla f_c(\mathbf{w}_t^i) - \nabla f(\mathbf{w}_t^i) \right), \quad \text{for } i = -2, -1 \tag{16}$$

which is smaller than the original drift in FedAvg, since $(1 - \alpha) < 1$. Thus, the magnitude of client drift for the last two layers is reduced, while the other layers are updated with standard SGD.

## 2.4 CONVERGENCE ANALYSIS

We provide a theoretical convergence analysis for both convex and non-convex settings in Appendix B.

## 3 EXPERIMENTS

### 3.1 DATASET

We utilized three well-established classification benchmarks: CIFAR10 (Krizhevsky et al., 2009), MNIST (LeCun et al., 2010), and FMNIST (Xiao et al., 2017). To make the non-IID data partitions, we replicated a partitioning strategy inspired by the approach detailed in (Lin et al., 2020), (Wang et al., 2020a), (Yurochkin et al., 2019). This involved distributing the data using a Dirichlet distribution with a concentration parameter $\beta$. The degree of data heterogeneity across clients is governed by the concentration parameter $\beta$ with a smaller value, resulting in a more skewed data distribution, mimicking real-world scenarios where data is unevenly partitioned. Figure 9 in the Appendix section demonstrates an example of the non-uniform data distribution observed in the MNIST dataset. In our experiments, we adopted $\beta$ values of $0.1$ and $0.3$, which are commonly employed values (Lin et al., 2020) to simulate varying levels of data heterogeneity. Each client possesses its own local data partition, which remains unchanged throughout the communication rounds. Refer to Appendix C.1 for the detailed experimental setup.

Table 1: Top-1 accuracy (%) on CIFAR10, MNIST, and FMNIST datasets with varying degrees of data heterogeneity. The values in bold represent the highest accuracy achieved. * represents the convergence failure by the respective algorithm.

| Dataset | CIFAR10 | | | | MNIST | | FMNIST | |
|---|---|---|---|---|---|---|---|---|
| Model | CNN | | VGG19 | | LeNet | | LeNet | |
| | $\beta = 0.1$ | $\beta = 0.3$ | $\beta = 0.1$ | $\beta = 0.3$ | $\beta = 0.1$ | $\beta = 0.3$ | $\beta = 0.1$ | $\beta = 0.3$ |
| Fedavg | 49.04±0.45 | 55.08±0.50 | 50.34±0.35 | 52.48±0.42 | 97.15±0.20 | 98.44±0.15 | 82.66±0.30 | 85.78±0.40 |
| FedProx | 50.38±0.48 | 55.12±0.44 | 47.01±0.33 | 52.39±0.39 | 97.65±0.22 | 98.62±0.18 | 82.33±0.28 | 85.97±0.38 |
| FedNova | 44.74±0.39 | 47.98±0.45 | 48.94±0.37 | 54.11±0.41 | 95.76±0.28 | 97.62±0.24 | 75.04±0.35 | 76.53±0.42 |
| FedBN | 51.84±0.41 | 54.62±0.43 | 10 (*) | 55.25±0.39 | 96.80±0.25 | 98.47±0.20 | 82.86±0.33 | 85.57±0.38 |
| FedDyn | 44.03±0.38 | 50.99±0.41 | 47.58±0.34 | 44.03±0.40 | 88.84±0.45 | 94.55±0.38 | 69.79±0.45 | 80.22±0.42 |
| MOON | 48.08±0.42 | 49.68±0.44 | 47.40±0.36 | 50.26±0.39 | 96.92±0.22 | 80.69±0.28 | 80.73±0.35 | 83.61±0.40 |
| SCAFFOLD | 50.34±0.44 | 55.76±0.47 | 52.13±0.40 | 56.29±0.42 | 97.19±0.21 | 98.46±0.19 | 81.75±0.32 | 85.09±0.39 |
| FedPVR | 47.04±0.40 | 49.89±0.43 | 49.05±0.38 | 42.93±0.37 | 97.34±0.22 | 98.46±0.18 | 80.95±0.34 | 82.32±0.40 |
| **FedPMVR** | **51.86**±0.45 | **56.07**±0.48 | **52.35**±0.41 | **56.41**±0.44 | **97.86**±0.20 | **98.67**±0.16 | **83.18**±0.33 | **86.14**±0.38 |

## 3.2 COMPARISON WITH THE STATE-OF-THE-ART METHODS

To assess the efficacy of the proposed FedPMVR, we selected eight popular FL algorithms for comparison: FedAvg (McMahan et al., 2017), FedProx (Li et al., 2020a), FedNova (Wang et al., 2020b), FedBN (Li et al., 2021b), FedDyn (Acar et al., 2021), MOON (Li et al., 2021a), SCAF-FOLD (Karimireddy et al., 2020), and FedPVR (Li et al., 2023). Our proposed FedPMVR demonstrated superior performance, achieving the highest top-1 accuracy across all experimental settings involving varying degrees of data heterogeneity ($\beta = 0.1$ and $0.3$) on three real-world datasets (refer to Table 1). Notably, in the highly heterogeneous setting ($\beta = 0.1$) on CIFAR10, FedP-MVR attained an accuracy of 51.86% with the CNN model and 56.07% with VGG19, surpassing the performance of the second-best methods, FedBN (51.84%) and Scaffold (55.76%), respectively. With $\beta = 0.3$, FedPMVR achieved accuracies of 56.07% and 56.41%, respectively, surpassing the runner-up method, SCAFFOLD, in both instances. Similarly, on MNIST with $\beta = 0.1$, FedPMVR reached 97.86% accuracy, exceeding FedProx (97.65%), and with $\beta = 0.3$, it accomplished 98.67% accuracy compared to FedProx's 98.62%. Lastly, on FMNIST with $\beta = 0.1$, FedPMVR attained 83.18% accuracy, outperforming FedBN (82.86%), and with $\beta = 0.3$, it achieved 86.14% accuracy, surpassing the second best FedProx (85.97%).

Table 2: Number of communication rounds required (speedup compared to FedAvg) to achieve the best top-1 accuracy of FedAvg. * represents the convergence failure by the respective algorithm.

| Dataset | CIFAR10 | | | | MNIST | | FMNIST | |
|---|---|---|---|---|---|---|---|---|
| Model | CNN | | VGG19 | | LeNet | | LeNet | |
| | $\alpha = 0.1$ | $\alpha = 0.3$ | $\alpha = 0.1$ | $\alpha = 0.3$ | $\alpha = 0.1$ | $\alpha = 0.3$ | $\alpha = 0.1$ | $\alpha = 0.3$ |
| Fedavg | 43 (1.0x) | 30 (1.0x) | 10 (1.0x) | 20 (1.0x) | 150 (1.0x) | 30 (1.0x) | 115 (1.0x) | 35 (1.0x) |
| FedProx | 40 (1.07x) | * | * | * | 60 (2.5) | 35 (0.85x) | 150 (0.77x) | 55 (0.67x) |
| FedNova | * | * | * | * | * | * | * | * |
| FedBN | * | 55 (0.54x) | * | 50 (0.4x) | * | 34 (0.88x) | 60 (1.92x) | 55 (0.67x) |
| FedDyn | * | * | * | * | * | * | * | * |
| MOON | 120 (0.35x) | * | * | * | * | * | * | * |
| SCAFFOLD | 27 (1.6x) | 25 (1.2x) | 10 (1.0x) | 22 (0.9x) | 85 (1.76x) | 23 (1.3x) | 60(1.92x) | 50 (0.7x) |
| FedPVR | * | * | 13 (0.77x) | 00 (0.0x) | **50 (3.0x)** | **23 1.3x)** | * | * |
| **FedPMVR** | **25 (1.7x)** | **21 (1.42x)** | **10 (1.0x)** | **20 (1.0x)** | **50 (3.0x)** | **23 (1.3x)** | **40 (2.88x)** | **30 (1.17x)** |

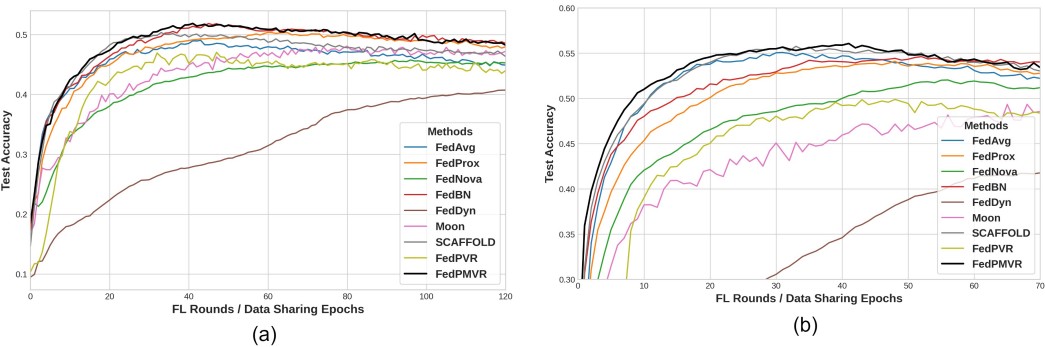

(a)  (b)

Figure 2: The performance comparison of proposed FedPMVR with baseline approaches using CNN model: (a) $\beta = 0.1$ and (b) $\beta = 0.3$ on the CIFAR10 dataset.

## 3.3 CONVERGENCE ANALYSIS

Following the methodology outlined in (Fan et al., 2022), we evaluated the number of communication rounds required by each approach to achieve the best accuracy attained by FedAvg. Table 2 presents the results for the required number of communication rounds and the corresponding speedup achieved by each algorithm. As evident from the results, certain algorithms, including MOON and FedPVR, failed to converge. In contrast, the proposed FedPMVR converged to the best accuracy attained by FedAvg while requiring the minimum number of communication rounds. Notably, FedPMVR achieves a speedup ranging from a minimum of 1.42 to a maximum of 3 times

compared to FedAvg across different non-IID settings and three real-world datasets. Additionally, FedPMVR does not necessitate any extra parameters, unlike SCAFFOLD and FedPVR. Consequently, FedPMVR operates equivalently to FedAvg in terms of the number of shared parameters while achieving superior test accuracy in fewer communication rounds. This efficiency and performance advantage make FedPMVR a compelling choice for FL tasks. This significant speedup underscores the superiority of our approach in terms of convergence speed and reduced communication overhead. Figure 2 shows the learning curves for CIFAR10, where test accuracy declines after a certain number of rounds for both $\beta$ values (0.1 and 0.3). Nonetheless, our proposed method consistently outperforms all baselines, achieving higher accuracy and demonstrating greater robustness to data heterogeneity. For the MNIST dataset (Fig. 3), several baseline methods fail to converge and exhibit divergence. In contrast, our proposed method consistently outperforms the baselines, surpassing them after 50 rounds (for $\beta = 0.1$) and 25 rounds (for $\beta = 0.3$), delivering consistently higher accuracy throughout the training process. A similar trend is observed for the FMNIST dataset (Fig. 4), where our method demonstrates superior stability during training while consistently outperforming the baseline approaches.

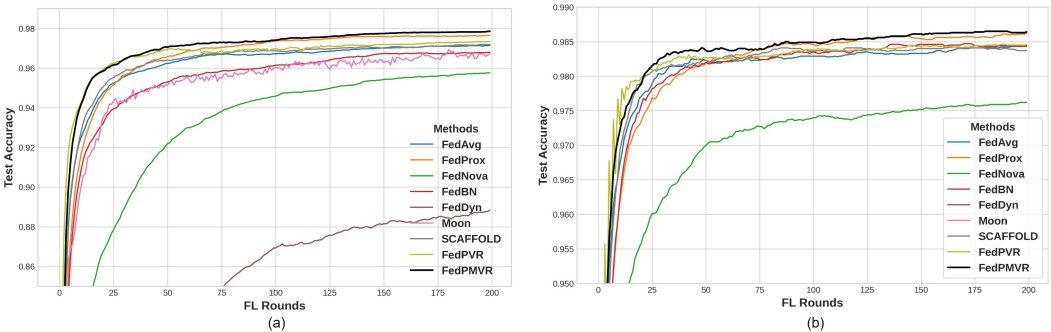

Figure 3: The performance comparison of proposed FedPMVR with baseline approaches: (a) $\beta = 0.1$ and (b) $\beta = 0.3$ on the MNIST dataset.

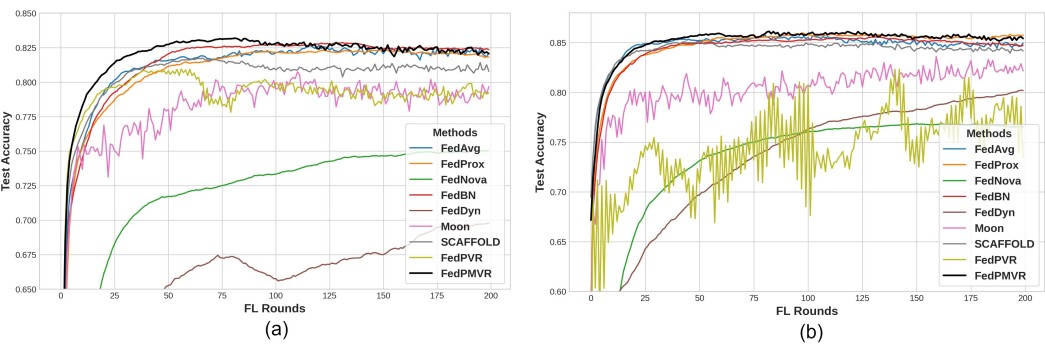

Figure 4: The performance comparison of proposed FedPMVR with baseline approaches: (a) $\beta = 0.1$ and (b) $\beta = 0.3$ on the FMNIST dataset.

## 3.4 CONFORMAL PREDICTION

Addressing significant data variability among clients in FL poses a significant challenge (Luo et al., 2021). To mitigate this issue, we employ a straightforward post-processing approach using conformal prediction to improve model performance. We analyze the empirical coverage and average size of the predictive set of the server model after 60 communication rounds. Empirical coverage measures the percentage of correct predictions within the predictive sets, while the average predictive set size represents the mean length of these sets across test images (Angelopoulos et al., 2022). Our

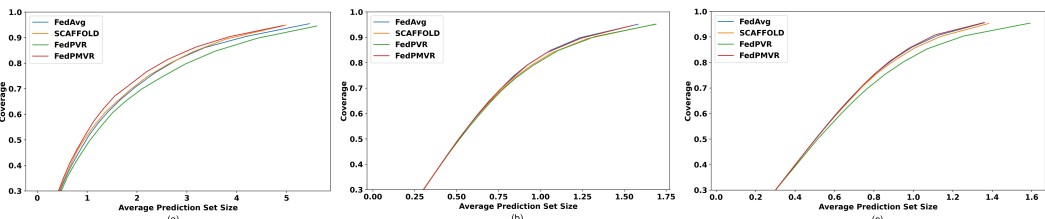

Figure 5: Relation between average predictive size and empirical coverage when $\beta = 0.1$.

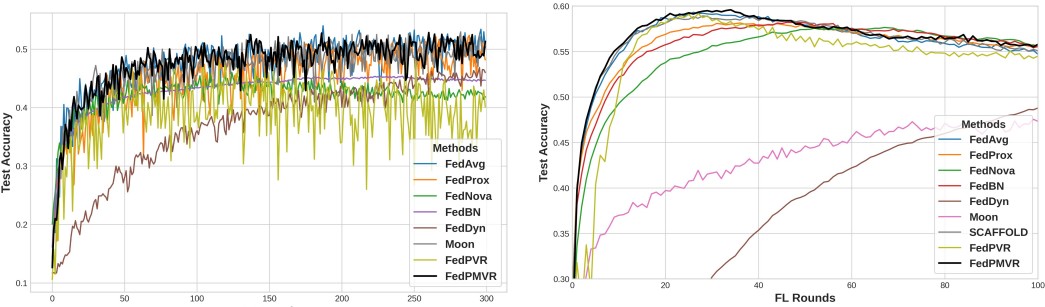

Figure 6: Experiments on partial participation of clients in the federated training process.

Figure 7: The performance comparison of proposed FedPMVR with baseline approaches on the IID Dataset..

findings reveal that a slight increase in the predictive set size can boost accuracy compared to individual models. Additionally, our approach often outperforms the top-1 accuracy of other individual models at a comparable or faster rate. Notably, as depicted in Fig. 5, the proposed model achieves similar or better coverage with smaller prediction sets than individual models, indicating the effective performance of the proposed model. In the coverage range between 0.8 and 0.9, our model maintains smaller prediction sets, demonstrating superior performance. Overall, the proposed model consistently achieves higher coverage with smaller prediction sets compared to individual models, effectively balancing high coverage with compact prediction sets, a desirable trait for practical applications.

### 3.5 PERFORMANCE ANALYSIS UNDER PARTIAL CLIENT PARTICIPATION

To align with more realistic FL scenarios, we conducted experiments using partial client participation, where only a subset of clients engaged in each FL round. Specifically, we simulated 100 clients, randomly selecting 20 (20% participation) per round. Using the CIFAR-10 dataset with $\beta = 0.1$ using VGG19 model, the results (shown in Fig. 6 and Table 3 in Appendix) demonstrate that FedPMVR achieves the highest accuracy of 51.33%, outperforming all baselines. Additionally, Fig. 6 highlights that FedPMVR not only achieves superior accuracy but also converges more quickly under this partial participation setting.

### 3.6 RESULTS ON IID DATASET

To evaluate the proposed FedPMVR method's efficacy on IID datasets, we constructed an IID partition of the CIFAR10 dataset by setting $\beta = 100$, used CNN network and compared its performance against other baselines. The results, presented in Fig. 7, demonstrate that FedPMVR outperforms the baseline methods and converges rapidly in the IID setting. This finding highlights that FedPMVR's applicability extends beyond non-IID data partitions, as it proves equally effective in IID data settings.

## 4 ABLATION STUDY

To gain deeper insights into the effectiveness of our method FedPMVR, we perform several experiments on the CIFAR10 dataset using CNN network in a highly heterogeneous setting ($\beta = 0.1$) unless explicitly stated otherwise.

### 4.1 EFFECT OF APPLYING MOMENTUM VARIANCE REDUCTION ON DIFFERENT LAYERS OF THE MODEL

To validate the rationale behind selectively applying our proposed momentum-based variance reduction techniques to the classification layers, we conducted experiments investigating the effects of applying this approach to other layers of the neural network model. Previous research (Li et al., 2023) highlights that client drift primarily manifests in the classification layers, underscoring the importance of addressing this issue in those layers. The results, depicted in Fig. 8, reveal that indiscriminately applying partial variance reduction across all layers adversely impacts performance. However, as we selectively applied partial variance reduction to the layers closer to the classifier, we achieved the highest accuracy and superior convergence rates. These findings corroborate our hypothesis and underscore the efficacy of our approach in mitigating client drift, specifically in the classification layers, which are most susceptible to this phenomenon. By judiciously targeting the classification layers with our momentum-based variance reduction techniques, we effectively address the root cause of client drift, that leads to improved model performance and convergence in FL settings.

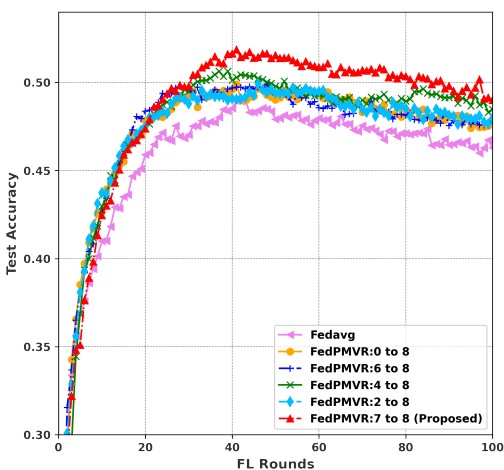

Figure 8: Effect of using momentum variance reduction on different layers in the neural network.

In addition to this experiment, we conducted two more experiments to investigate the impact of the hyperparameter $\alpha$ on test accuracy and the performance of the proposed model with the baselines under larger client participation. These detailed results can be found in subsections C.2 and C.3 in the Appendix.

## 5 CONCLUSION

In this paper, we investigated data heterogeneity within the traditional FL framework, focusing on enhancing the accuracy and convergence speed of the final global model. We conducted an in-depth analysis of client drift in FL, which arises from data heterogeneity, causing local models to diverge from the global model and reducing the effectiveness of aggregated updates. To address this, we introduced FedPMVR, which employs a straightforward yet effective partial momentum variance reduction technique to stabilize training, accelerate convergence, and improve overall global model performance. The core advantage of FedPMVR lies in its ability to mitigate client drift through momentum-based variance reduction, ensuring stable gradient updates without increasing much communication overhead. Our comprehensive experiments across multiple datasets confirm FedPMVR's superior performance, particularly in highly non-IID data scenarios, demonstrating significant improvements in both convergence speed and model accuracy. Importantly, FedPMVR requires only a minor modification to the existing FedAvg, offering an efficient and scalable solution for real-world FL applications. Additionally, while FedPMVR introduces minimal computational overhead at the client side for calculating the momentum term, future work will focus on exploring strategies to further mitigate this cost. We also aim to extend our approach to more complex scenarios, such as asynchronous updates and the integration of advanced privacy-preserving techniques.

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

## APPENDIX

## A RELATED WORK

Various approaches have been extensively explored to address the challenges arising from data heterogeneity in the context of FL. These methodologies can be broadly classified into three main categories: (1) client drift mitigation strategies (Li et al., 2021a), (Karimireddy et al., 2020), (Sahoo et al., 2024b), (Li et al., 2023) (Sahoo et al., 2024a), which modify the local objectives of clients to better align their models with the global model; (2) aggregation scheme enhancements (Hsu et al., 2019), (Lin et al., 2020), (Wang et al., 2020b), (Wang et al., 2020a), which focus on improving the server-side fusion mechanism for model updates; and (3) personalized FL techniques (Fallah et al., 2020), (Sattler et al., 2020), (Bui et al., 2019), which aim to train personalized models for individual clients. Since our work centers on minimizing client-server model divergence to mitigate client drift, we primarily discuss client drift mitigation strategies.

FedAvg (McMahan et al., 2017) is the most widely adopted optimization method in FL, but data heterogeneity often results in subpar performance. To tackle this issue, methods like FedProx (Li et al., 2020a) have introduced a proximal regularization term at the client side to manage divergence; however, it still falls short in effectively aligning the optimal points (Acar et al., 2021). FedNova (Wang et al., 2020b) tackled the issue of objective inconsistency caused by client heterogeneity in federated optimization by introducing a normalized averaging method, which effectively mitigates this inconsistency and accelerates error convergence. Li et al. (2021b) used local batch normalization to mitigate feature shifts, and demonstrated faster convergence. Acar et al. (2021) proposed a dynamic regularizer aimed at bridging the gap between local and global minima, facilitating better alignment of local solutions with the global model. Li et al. (2021a) introduced a model-contrastive

framework that aligns local representations with the global model using contrastive loss, enhancing consistency and convergence, particularly in non-IID settings. Similarly, FedMut (Hu et al., 2024) generates intermediate models through global model mutation, which avoids sharp solutions, improves generalization in heterogeneous data, and outperforms FedAvg in handling data heterogeneity. Stochastic variance reduction (SVR) techniques, such as SVRG (Johnson & Zhang, 2013), SAGA (Defazio et al., 2014), and their variants, have been explored to mitigate challenges posed by data heterogeneity. These methods leverage control variate to reduce the variance inherent in traditional stochastic gradient descent (SGD), enabling linear convergence rates for strongly convex optimization problems. SCAFFOLD (Karimireddy et al., 2020) and DANE (Shamir et al., 2014) have incorporated variance reduction techniques for the entire model on convex problems, but their performance in non-convex setups has not been extensively explored. While these approaches have the potential to be beneficial, they incur higher communication costs due to the transmission of additional control variates (Halgamuge et al., 2009). Moreover, existing methods have demonstrated rapid convergence in simpler models, but their effectiveness on deep neural networks (DNNs) (He et al., 2015), (Huang et al., 2018) remains largely unexplored. FedPVR (Li et al., 2023) revisits the performance of FedAvg in DNNs and reveals significant diversity in the final classification layers. By proposing variance reduction solely on the final layers, FedPVR outperforms several benchmarks, addressing the limitations of existing approaches and demonstrating the effectiveness of targeted variance reduction techniques. Several momentum-based techniques have been explored to enhance FL convergence (Das et al., 2022). Liu et al. (2020) incorporated Momentum Gradient Descent (MGD) into the local update step, accelerating convergence, deriving an upper bound on the convergence rate, and identifying conditions where it outperforms standard FL. Building on this, Cheng et al. (2023) demonstrated that momentum improves FedAvg and Scaffold, allowing FedAvg to converge without assuming bounded data heterogeneity, even with a constant local learning rate. They also showed that momentum accelerates Scaffold's convergence under partial client participation, leading to new variance-reduced extensions with state-of-the-art convergence rates. Further, Sun et al. (2024) introduced a general framework for server momentum in FL, accommodating diverse momentum schemes, stagewise hyperparameter scheduling, and handling system heterogeneity and asynchronous local computing.

# B  THEORETICAL CONVERGENCE ANALYSIS

This section will provide a theoretical analysis of the FedPMVR in both convex and non-convex settings. We begin by establishing assumptions that are akin to those employed in the FedAvg algorithm (McMahan et al., 2017), (Li et al., 2020b), followed by discussing convergence guarantees.

## ASSUMPTIONS AND NOTATION

- **Convexity:** The local objective function for client $i$ $f_i$ is convex, i.e., $f_i(y) \geq f_i(x) + \langle \nabla f_i(x), y - x \rangle$ for all $x, y$.

- **Lipschitz Smoothness:** $|\nabla f_i(x) - \nabla f_i(y)| \leq L|x - y|$ for all $x, y$ and some constant $L > 0$.

- **Bounded Gradients:** $\mathbb{E}|\nabla f_i(x; \xi_i)|^2 \leq G^2$ for all $x$ and some constant $G > 0$, where $\xi_i$ denotes the random variable representing the data samples used to compute stochastic gradients on client $i$.

- **Data Heterogeneity:** Similar to the (Li et al., 2023), We assume there exists constant $\hat{\zeta}$ such that $\forall x \in R^d$: $\frac{1}{N} \sum_{i=1}^{N} \mathbb{E}|\nabla f_i(x)|^2 \leq \hat{\zeta}^2, \quad \forall x.$

## B.1  CONVEX SETTING

Using the convexity property of $F$:

$$F(y_i^{(r,k+1)}) \leq F(y_i^{(r,k)}) + \langle \nabla F(y_i^{(r,k)}), y_i^{(r,k+1)} - y_i^{(r,k)} \rangle, \tag{17}$$

where $F$ is the global objective function and $y_i^{(r,k)}$ is the local model parameters for client $i$ at round $r$ and iteration $k$. Substituting the local update with momentum (Eq. 18) with Eq. 17, we obtain Eq. 19.

$$y_i^{(r,k+1)} = y_i^{(r,k)} - \eta_l(g_i^{(r,k)} + m_i^{(r,k)}), \tag{18}$$

where $\eta_l$ is the local learning rate, $g_i^{(r,k)}$ is the local stochastic gradient for client $i$ at round $r$ and iteration $k$, and $m_i^{(r,k)}$ is the momentum term for client $i$ at round $r$ and iteration $k$.

$$F(y_i^{(r,k+1)}) \le F(y_i^{(r,k)}) - \eta_l \langle \nabla F(y_i^{(r,k)}), g_i^{(r,k)} + m_i^{(r,k)} \rangle. \tag{19}$$

Taking the expectation (Eq. 19) with respect to the stochastic gradients, we obtain Eq. 20.

$$\mathbb{E}[F(y_i^{(r,k+1)})] \le \mathbb{E}[F(y_i^{(r,k)})] - \eta_l \mathbb{E}[\langle \nabla F(y_i^{(r,k)}), \nabla f_i(y_i^{(r,k)}) \rangle]. \tag{20}$$

Using $\mathbb{E}[\langle \nabla F(y_i^{(r,k)}), \nabla f_i(y_i^{(r,k)}) \rangle] = \|\nabla F(y_i^{(r,k)})\|^2$ in the Eq. 20, we obtain Eq. 21.

$$\mathbb{E}[F(y_i^{(r,k+1)})] \le \mathbb{E}[F(y_i^{(r,k)})] - \eta_l \|\nabla F(y_i^{(r,k)})\|^2. \tag{21}$$

Summing over $K$ local updates, we obtain Eq. 22:

$$\mathbb{E}[F(y_i^{(r,K)})] \le \mathbb{E}[F(y_i^{(r,0)})] - \eta_l \sum_{k=0}^{K-1} \|\nabla F(y_i^{(r,k)})\|^2. \tag{22}$$

We take the average over clients and obtain Eq. 23:

$$\frac{1}{N} \sum_{i=1}^{N} \mathbb{E}[F(y_i^{(r,K)})] \le \frac{1}{N} \sum_{i=1}^{N} \mathbb{E}[F(y_i^{(r,0)})] - \eta_l \sum_{k=0}^{K-1} \frac{1}{N} \sum_{i=1}^{N} \|\nabla F(y_i^{(r,k)})\|^2. \tag{23}$$

Using the global model update (Eq. 24) in the Eq. 23 and taking the average over $R$ communication rounds, we obtain Eq. 25:

$$x^{(r+1)} = \frac{1}{N} \sum_{i=1}^{N} y_i^{(r,K)}, \tag{24}$$

where $x^{(r)}$ is the global model parameters at round $r$.

$$\frac{1}{R} \sum_{r=0}^{R-1} \mathbb{E}[F(x^{(r)}) - F^*] \le \mathcal{O}\left( \frac{G\sqrt{K}}{\sqrt{NR}} + \frac{\hat{\zeta}\sqrt{K}}{\sqrt{R}} + \frac{\hat{\zeta}_p\sqrt{K}}{\sqrt{NR}} + \frac{F(x^{(0)}) - F^*}{R} \right) \tag{25}$$

where $F^*$ is the optimal value of the global objective function $F$, and $K$ is the number of local updates. The convergence rate is influenced by the number of communication rounds $R$, the number of clients $N$, the gradient bound $G$, and data heterogeneity $\hat{\zeta}$. As $R$ increases, the algorithm converges faster.

## B.2 NON-CONVEX SETTING

Given the smoothness property of $F$, we can get Eq. 26:

$$F(y_i^{(r,k+1)}) \le F(y_i^{(r,k)}) + \langle \nabla F(y_i^{(r,k)}), y_i^{(r,k+1)} - y_i^{(r,k)} \rangle + \frac{L}{2} \|y_i^{(r,k+1)} - y_i^{(r,k)}\|^2. \tag{26}$$

Substituting the local update with momentum (Eq. 27) in the Eq. 26, we obtain Eq. 28:

$$y_i^{(r,k+1)} = y_i^{(r,k)} - \eta_l(g_i^{(r,k)} + m_i^{(r,k)}) \tag{27}$$

$$F(y_i^{(r,k+1)}) \le F(y_i^{(r,k)}) - \eta_l \langle \nabla F(y_i^{(r,k)}), g_i^{(r,k)} + m_i^{(r,k)} \rangle + \frac{L}{2} \eta_l^2 \|g_i^{(r,k)} + m_i^{(r,k)}\|^2. \tag{28}$$

Taking the expectation (Eq. 28) with respect to the stochastic gradients, we obtain Eq. 29:

$$\mathbb{E}[F(y_i^{(r,k+1)})] \le \mathbb{E}[F(y_i^{(r,k)})] - \eta_l \mathbb{E}[\langle \nabla F(y_i^{(r,k)}), \nabla f_i(y_i^{(r,k)}) \rangle] + \frac{L}{2} \eta_l^2 \mathbb{E}[\|\nabla f_i(y_i^{(r,k)}) + m_i^{(r,k)}\|^2]. \tag{29}$$

Using $\mathbb{E}[\langle \nabla F(y_i^{(r,k)}), \nabla f_i(y_i^{(r,k)})\rangle] = \|\nabla F(y_i^{(r,k)})\|^2$, in Eq. 29, we obtain Eq. 30:

$$\mathbb{E}[F(y_i^{(r,k+1)})] \le \mathbb{E}[F(y_i^{(r,k)})] - \eta_l\|\nabla F(y_i^{(r,k)})\|^2 + \frac{L}{2}\eta_l^2\mathbb{E}[\|\nabla f_i(y_i^{(r,k)}) + m_i^{(r,k)}\|^2]. \quad (30)$$

Using $\mathbb{E}[\|\nabla f_i(y_i^{(r,k)})\|^2] \le G^2$, in Eq. 30, we obtain Eq. 31:

$$\mathbb{E}[\|\nabla f_i(y_i^{(r,k)}) + m_i^{(r,k)}\|^2] \le 2\mathbb{E}[\|\nabla f_i(y_i^{(r,k)})\|^2] + 2\mathbb{E}[\|m_i^{(r,k)}\|^2] \le 2G^2 + 2\|m_i^{(r,k)}\|^2 \quad (31)$$

Combining Eq. 30 and Eq. 31, we obtain Eq. 32:

$$\mathbb{E}[F(y_i^{(r,k+1)})] \le \mathbb{E}[F(y_i^{(r,k)})] - \eta_l\|\nabla F(y_i^{(r,k)})\|^2 + L\eta_l^2(G^2 + \|m_i^{(r,k)}\|^2). \quad (32)$$

Summing over $K$ local updates and $R$ communication rounds and averaging over clients, we obtain Eq. 33:

$$\frac{1}{R}\sum_{r=0}^{R-1}\mathbb{E}[F(x^{(r)}) - F^*] \le \mathcal{O}\left(\frac{G\sqrt{K}}{\sqrt{NR}} + \frac{\hat{\zeta}K^{3/2}}{\sqrt{R}} + \frac{\hat{\zeta}_p\sqrt{K}}{\sqrt{N}} + \frac{F(x^{(0)}) - F^*}{R}\right), \quad (33)$$

where $\hat{\zeta}_p$ is the heterogeneity measure for the layers with applied momentum correction. The convergence rate also depends on the number of local updates $K$ and the heterogeneity measure $\hat{\zeta}_p$. The presence of $\hat{\zeta}_p$ indicates that applying momentum correction to the last layers can potentially improve the convergence rate if the gradients for these layers are more aligned across clients.

---

**Algorithm 1** FedPMVR

---

1: **Server Initialization:** Initialize the global model weights. $\mathbf{W}_0$
2: **Client Initialization:** Initialize the clients momentum terms $\mathbf{m}_0^i = \mathbf{0}$ for all the clients, learning rate $\alpha$ for momentum update, the number of local epochs $E$ and the local learning rate $\eta$.
3: Define the mask $\mathbf{p} \in \{0,1\}^d$ with $v$ non-zero elements (indices of last few layers).
4: $\mathcal{S}_{pvr} \leftarrow \{j : p_j = 1\}$ ▷ Indices of layers for momentum correction
5: $\mathcal{S}_{sgd} \leftarrow \{j : p_j = 0\}$ ▷ Indices of layers without momentum correction
6: **Client Update:**
7: **for** each communication round $t = 1, 2, \ldots, T$ **do**
8:     Broadcast the global model weights $\mathbf{W}_r$ to all the clients.
9:     **for** each client $c \in \{1, 2, \ldots, C\}$ **in parallel do**
10:         Initialize the local model weights $\mathbf{w}_c \leftarrow \mathbf{W}_t$
11:         Initialize the client momentum terms $\mathbf{m}_c^{client} \leftarrow \mathbf{0}$
12:         **for** $e = 1, 2, \ldots, E$ **do**
13:             Update the local model weights $\mathbf{w}_k$ using SGD on the client's local data.
14:         **end for**
15:         Compute the gradients $\mathbf{g}_c = \nabla_{\mathbf{w}}\mathcal{L}(\mathbf{w}_c; \mathcal{D}_c)$.
16:         **for** $j \in \mathcal{S}_{pvr}$ **do**
17:             Update the client momentum term: $m_{c,j}^{client} \leftarrow \alpha g_{c,j} + (1-\alpha)m_{c,j}^{client}$
18:             Correct the weight: $w_{c,j} \leftarrow w_{c,j} - m_{c,j}^{client}$
19:         **end for**
20:         **for** $j \in \mathcal{S}_{sgd}$ **do**
21:             Update the weight: $w_{c,j} \leftarrow w_{c,j} - \eta g_{c,j}$
22:         **end for**
23:         Send the corrected local model weights $\mathbf{w}_c$ to the server
24:     **end for**
25: **end for**
26: **Server Aggregation:**
27: Aggregate the received local model weights: $\mathbf{w}_{t+1} \leftarrow \frac{1}{C}\sum_{c=1}^{C}\mathbf{w}_t^c$

---

## C  EXPERIMENTAL DETAILS

### C.1  EXPERIMENTAL SETUP

For our experiments with MNIST and FMNIST, we have utilized the well-established LeNet (LeCun et al., 1998) neural network, commonly employed in FL research (Duan et al., 2023). For the CIFAR10 dataset, we constructed a custom 5-layer convolutional neural network (CNN) architecture from scratch to serve as the base encoder, following the approach presented in (Li et al., 2021a). This base encoder composed of two convolutional layers with $(5 \times 5)$ kernels, each succeeded by a $(2 \times 2)$ max pooling layer. Subsequently, the architecture includes two fully connected (FC) layers with ReLU activation functions, the first containing 120 and the second containing 84 units of neurons. Following the methodology outlined in (Duan et al., 2023), we applied variance reduction techniques to the last two layers of the model to tackle the impact of data heterogeneity. We followed the configurations reported in prior studies (Li et al., 2023) and (Yu et al., 2022) by involving 10 participating clients per communication round and employing a batch size of 32. Each local client performed two local epochs of model updating during each round. We have employed a maximum of 300 server-client communication rounds with the early stopping criteria. To evaluate the global model's performance, a test set, unseen during training, is held out at the server. We utilize the actual test split from the original dataset for this purpose. To determine the optimal client learning rate for each experiment, we conducted a grid search over $0.05, 0.01, 0.2, 0.3$. For all datasets and methods, the optimizer used was SGD with a learning rate of $0.01$, a weight decay of $1e-6$, and a momentum value of $0.9$. we conducted a grid search over $0.001, 0.1, 0.3, 0.6, 0.8$ to choose the learning rate for momentum update $\alpha$. We employed hyperparameters (if any) similar to those specified in the original papers for all baseline methods. We have run each experiment three times with different seed values and reported the average of the performance with the standard deviation.

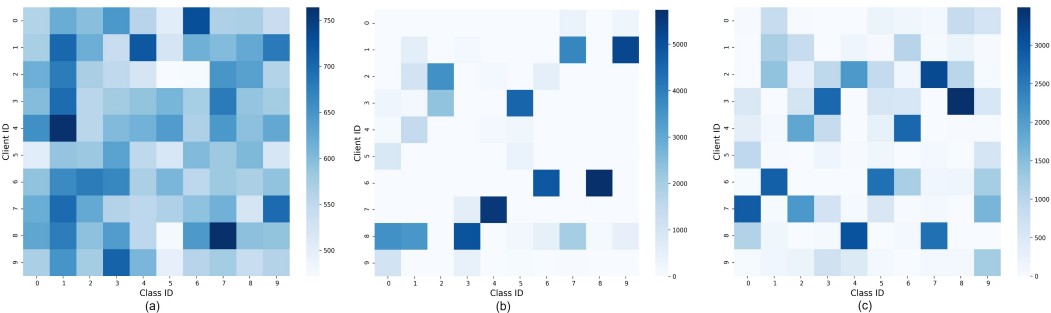

Figure 9: Visualization of the data distribution among local clients for MNIST using Dirichlet sampling under (a) IID ($\beta = 100$), (b) non-IID ($\beta = 0.1$), and (c) non-IID ($\beta = 0.3$) conditions. Each grid cell represents the number of samples of a particular class assigned to a client. Darker colors indicate a higher number of samples allocated to the client.

Table 3: Performance comparison of FedPMVR and baseline methods on the CIFAR10 dataset with $\beta = 0.1$ using the VGG19 model under partial client participation.

| Method | Accuracy |
|---|---|
| FedAvg | 50.21 ± 0.35 |
| FedProx | 48.17 ± 0.40 |
| FedNova | 41.87 ± 0.50 |
| FedBN | 44.66 ± 0.30 |
| FedDyn | 45.99 ± 0.45 |
| MOON | 50.96 ± 0.33 |
| SCAFFOLD | 10 (*) |
| FedPVR | 40.11 ± 0.55 |
| **FedPMVR (Proposed)** | **51.33 ± 0.28** |

918
919

## C.2 EFFECT ON TEST ACCURACY FOR DIFFERENT VALUES OF HYPERPARAMETER $\alpha$

920
921
922
923
924
925
926
927
928
929

The hyperparameter $\alpha$ controls the degree of variance reduction in the classification layers by modulating the momentum update. $\alpha$ can take values between 0 and 1. To gain insights into the nature of this hyperparameter, we experimented with a range of values from lower to higher magnitudes. As depicted in Fig. 10, higher values of $\alpha$ lead to lower accuracy, while values closer to zero yields higher accuracy. A possible explanation for this observation is that when $\alpha$ is set to a higher value, instead of mitigating client drift, it may inadvertently exacerbate drift by aggressively reducing the weight magnitudes in an attempt to align with the global model for achieving the global optimum. Conversely, lower values of $\alpha$ closer to zero effectively curb client drift, resulting in improved accuracy. These findings suggest that $\alpha$ is an easily tunable hyperparameter, with lower values close to zero proving to be the most effective in enhancing the performance of our proposed approach.

930
931
932
933
934
935
936
937
938
939
940
941
942
943
944
945
946
947
948
949
950

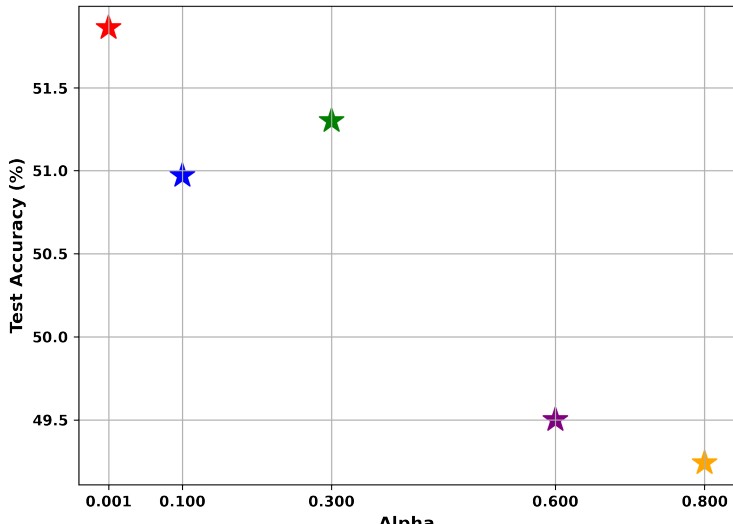

951
Figure 10: Effect of different values of learning rate for momentum update $\alpha$.

952
953
954

## C.3 PERFORMANCE ANALYSIS IN LARGE-SCALE CLIENT SETTINGS

955
956
957
958
959
960
961
962
963

To assess the scalability, we evaluated the performance of FedPMVR and baselines by varying the number of participating clients, as depicted in Fig. 11. As the number of clients increased from 10 to 50, FedPMVR exhibited a significantly slower decline in accuracy compared to the baselines. This observation demonstrates the superior scalability of our method, enabling it to maintain high accuracy even as the number of participating clients grows. Notably, FedPMVR achieved the highest test accuracy among all baselines in all evaluated cases, further underscoring its robust performance and ability to handle increasing client populations effectively. This scalability is a desirable attribute, particularly in real-world FL scenarios involving large numbers of clients, where our method can deliver consistent and reliable results.

964
965

## C.4 LIMITATIONS

966
967
968
969
970
971

While this work primarily investigates data heterogeneity in terms of label and quantity skewness, the concept of momentum-based variance reduction could be expanded to address other forms of heterogeneity, such as feature-level heterogeneity in classification tasks. Additionally, the implementation of FedPMVR incurs some computational overhead due to the use of momentum terms for last-layer updates on the client side, an issue we plan to investigate further in future work to mitigate this impact. Furthermore, despite federated learning's advantages in privacy preservation and collaborative learning, it remains vulnerable to privacy leakage from malicious clients, as highlighted

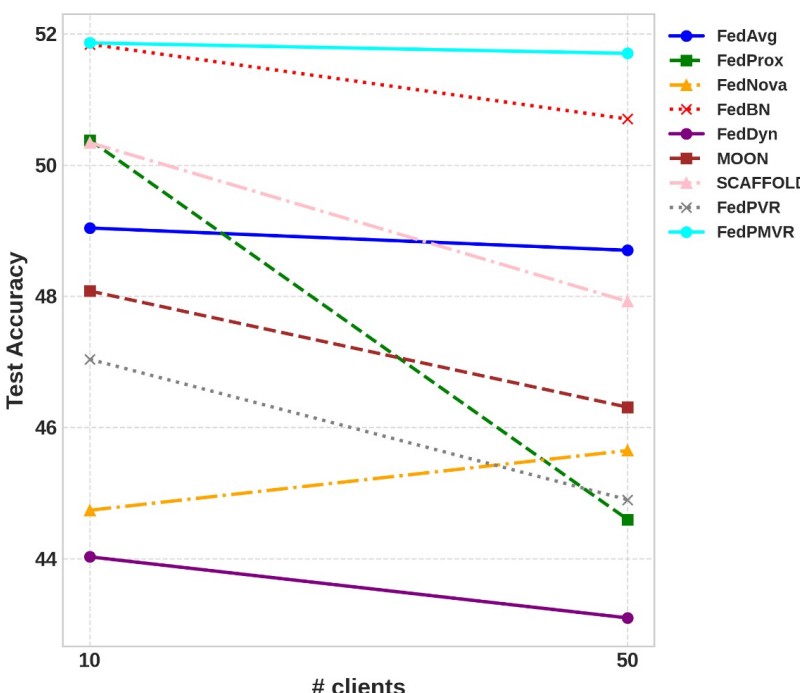

Figure 11: Test accuracy of the proposed and baseline algorithms with different numbers of clients.

in recent studies (Zhang et al., 2022), (Cao et al., 2021). This paper does not delve into this aspect and considers it a potential avenue for future research.

