# OpenReview forum: "FedPMVR: Addressing Data Heterogeneity in Federated Learning through Partial Momentum Variance Reduction"
_ICLR.cc/2025/Conference — ICLR 2025 Conference Withdrawn Submission_

### Official Review · Reviewer_NQdy · 2024-10-26

**Soundness:** 2
**Presentation:** 2
**Contribution:** 2
**Rating:** 5
**Confidence:** 4

**Summary:**

The paper introduces a novel method, FedPMVR, which integrates momentum-based partial variance reduction (VR) to address data heterogeneity in federated learning (FL) settings. Specifically, the method aims to mitigate the negative effects of non-iid (independent and identically distributed) data across clients, often leading to drift and slow convergence. FedPMVR selectively applies momentum to the classification layers of client models while relying on standard stochastic gradient descent (SGD) for other layers. The authors claim that this technique improves global model accuracy and faster convergence. The method is evaluated on popular benchmark datasets (CIFAR-10, MNIST, and FMNIST) under varying degrees of data heterogeneity, and results show performance improvements over state-of-the-art methods.

**Strengths:**

**Solid Empirical Results**: The authors provide thorough experimental results on widely-used datasets with varying heterogeneity levels, demonstrating consistent performance improvements over existing FL methods such as FedAvg, FedProx, and SCAFFOLD. The accuracy gains across datasets and models are significant and support the effectiveness of the proposed method.

**Comprehensive Literature Review**: The paper does a commendable job of situating its work within the broader federated learning landscape. The authors demonstrate a deep understanding of the field by discussing and comparing FedPMVR with a wide array of contemporary methods (e.g., FedAvg, FedProx, MOON, SCAFFOLD, and FedPVR). This thorough review of related works strengthens the paper by highlighting gaps in the existing literature that FedPMVR aims to fill.

**Theoretical Guarantees**: The paper includes a convergence analysis for both convex and non-convex settings, which adds rigor to the empirical claims and ensures that the method is theoretically sound.

**Weaknesses:**

**Clarity and Structure**: The method's presentation is somewhat convoluted. While the paper discusses client drift and the theoretical aspects in detail, the specific steps of the proposed algorithm are not as clearly articulated. More emphasis should be placed on the step-by-step explanation of the FedPMVR method, and the illustrations, though helpful, could be clearer in communicating the concepts.

**Limited Scope of Non-IID Analysis**: The paper assumes specific levels of non-IIDness by using predefined parameters for data heterogeneity. However, the method's broader applicability under different types of client heterogeneity (e.g., feature shift, label distribution skew) is not explored in detail. Further improvement in removing the heterogeneity condition could strengthen the paper.

**Computational Overhead**: While the paper claims that the method introduces minimal computational overhead at the client level, this is not quantified meaningfully. Since the method involves momentum updates for some layers, a deeper analysis of the impact on client-side computation and memory would be beneficial, especially in resource-constrained settings.

**Questions:**

What is the additional computational burden on client devices when applying momentum-based updates, especially for large-scale models with deeper classification layers?

---

### Official Review · Reviewer_zFeF · 2024-11-01

**Soundness:** 2
**Presentation:** 2
**Contribution:** 2
**Rating:** 3
**Confidence:** 4

**Summary:**

This paper deals with data heterogeneity in Federated learning. Data heterogeneity is a common problem in FL. This paper presents a method called FedPMVR, which performs partial momentum-based variance reduction on the client side. The authors demonstrate their approach with extensive experimental results and achieve better performance than several existing methods across various datasets and deep neural networks.

**Strengths:**

The authors presented extensive experimental results across three datasets—CIFAR-10, MNIST, and FMNIST—various levels of data heterogeneity, and different neural networks.

**Weaknesses:**

- This paper proposes the idea of partial momentum-based variance reduction, which combines the approaches of FedPVR and MIME. MIME introduces momentum-based variance reduction, while FedPVR demonstrates partial variance reduction. The authors are encouraged to present more detailed comparisons against these two works.

- This paper presents extensive experimental results using 10 clients and neural networks such as CNN, VGG19, and LeNet. The authors are encouraged to explain their choice of these networks. Additionally, will the results generalize well to more complex networks, such as ResNet or vision transformers? Will the results also generalize to a higher number of clients?

- I notice that the experimental results section has a structure very similar to that of FedPVR, covering topics such as experimental results demonstration, convergence rate analysis, conformal prediction, and the effect of partial variance reduction on different layers. The authors are encouraged to explain the rationale behind this structure.


References:
- MIME: Mime: Mimicking Centralized Stochastic Algorithms in Federated Learning, Karimireddy et al., 2021
- FedPVR: On the effectiveness of partial variance reduction in federated learning with heterogeneous data, Li et al., 2023

**Questions:**

- In Table 1, I was surprised to see that FedAvg performs much better than more advanced methods, such as FedDyn and FedPVR, when the clients are heterogeneous ($\beta$=0.1 and $\beta$=0.3), across nearly all datasets and neural networks. Could the authors elaborate on the strong performance of FedAvg?

- In Figures 2 and 7, test accuracy is already decreasing over FL rounds. What is the reason for not implementing early stopping? Does the top-1 accuracy presented in Table 1 correspond to the accuracy at round 120?

- In Figure 5, what are the differences among a, b, and c? Why does each method have a different endpoint on the x-axis?

---

### Official Review · Reviewer_YGxC · 2024-11-04

**Soundness:** 1
**Presentation:** 1
**Contribution:** 1
**Rating:** 3
**Confidence:** 4

**Summary:**

The author proposed a partial variance reduction method to mitigate the client drift problem in federated learning.
Client drift is caused by data heterogeneity, which is unavoidable due to the main assumption of federated settings.
Since the gradient dissimilarity is prominent in the penultimate layer, the momentum term is selectively assigned to this layer to reduce the variability between the last layer of local and global models. The effectiveness of the proposed method, `FedPMVR` is evaluated three image classification benchmark datasets.

**Strengths:**

## S1. Significance of the Target Problem
---
- The authors first proposed a partial momentum method to reduce the negative effect of client drift in federated learning.
- The client drift problem is one of the core issues to be solved for the success of the federated system, since it often hinders the convergence of a global model, or even the failure.

## S2. Novelty
---
- Client drift can be quantified in various ways; the authors tried to model it using the discrepancy between the global optimum and the local models in the parameter space.
- Based on the observation that the discrepancy is especially pronounced in the near-final layers, the authors proposed to reduce the gap by tracking the local momentum term and smoothing the unavoidable differences.

**Weaknesses:**

## W1: Justification of Main Argument
---
- In the absence of an in-depth comparison with the most similar previous work [1], the originality of the proposed method is inevitably accepted as incremental. For example, Section 4.1 overlaps significantly with Figure 5 of [1] in terms of illustration and message.
- The justification of the authors' main claim on their proposed momentum term is neither empirically nor theoretically sufficient, e.g., on the stabilization of the gradient updates (lines 219-220), and the acceleration in convergence (lines 222-224).
- The convergence analyses in section 2.4 and section B have much room for improvement. I can never find the effect of the proposed momentum, quantified by $\alpha$ in both eq. (25) and eq. (33). Additionally, the intermediate steps of each proof are overly omitted, making it difficult to understand the theoretical findings. Thus, it is unclear exactly what 'variance' is reduced due to the proposed method.
- It seems that the analyses are mostly adopted from [1], but it is neither cited nor adopted correctly in Section B.

## W2. Empirical Validation
---
- The scale of the experiment is unsatisfactory for a FL algorithm. There are only 10 participating clients in the system, no (local) learning rate decay, which is critical for FL algorithm to be converged[2]. Plus, the experiments are limited only to small-scale image recognition tasks. The authors can refer to widely used benchmarks such as LEAF[3], FLamby[4], StackOverflow[5], and Google Landmark[6].
- Although the authors proposed local momentum method to mitigate client drift, the evaluation is done on central dataset (line 881), not on local evaluation sets. This discrepancy is not convincing to validate the effectiveness of `FedPMVR`.
- While the authors proposed to use the partial momentum assuming that there are two final layers (line 273), there can be only a single final layer in some readymade model architectures. In this context, I think the effect of the number of final layers should also be studied in Section 4.

## W3: Overall Presentation
---
- The notation should be clearly defined with its correct name, symbol and dimension, and be used consistently throughout the paper.
- While the authors focused on the problem of client drift in federated learning, the related work presented in the paper is too broad for the reader to easily understand the stream of similar research (e.g., regularization of (partial) layers or model of each client).

[1] On the effectiveness of partial variance reduction in federated learning with heterogeneous data (2022)
[2] On the Convergence of FedAvg on Non-IID Data (2020)
[3] LEAF: A Benchmark for Federated Settings (2019)
[4] FLamby: Datasets and Benchmarks for Cross-Silo Federated Learning in Realistic Healthcare Settings (2022)
[5] https://www.tensorflow.org/federated/api_docs/python/tff/simulation/datasets/stackoverflow/load_data (TensorFlow Federated)
[6] Federated Visual Classification with Real-World Data Distribution (2020)

**Questions:**

## Q1. Line 103 (`With increased heterogeneity (lower $\beta$)')
---
* The notation $\beta$ is first used here, but I can find the definition from nowhere. Where is it defined?
* From the following sentence, it seems an important symbol related to client drift, the main target to be tackled in this research; it should be clearly defined.

## Q2. Line 110-169 (Case studies on data heterogeneity and client drift)
---
* The argument from two case studies is too bold and does not fully support the claim. What if the $\eta$ is chosen to be $\eta=3/4.75$?
* In this case, the eq. (4) in Case 2 (high heterogeneity case) becomes $w^1=3=W^\star$, while the eq. (3) in Case 1 becomes $w^1 \approx 10.7$, which is far more distant from the global optimum of $W^\star=3$.

## Q3. Line 206-215 and Line 48-62 & 210 in `FedPMVR.py` of supplementary material (code)
---
* Different from eq. (7), the local sample size ($n_c$) is ignored in the real implementation. Please clarify the gap between the description and implementation.
* Note that the local sample sizes are usually incorporated in plenty of FL literature to be aligned with the empirical risk minimization principle.

## Q4. Line 206-266
---
* It is questionable why client index is changed from $i$ to $c$ in this section, apart from the first definition in section 2.1. If there is no evident reason, please make sure to be consistent in writing notations.

## Q5. Line 206-215 and Line 236-256
---
* Is there any difference between $\mathbf{W}$ and $\mathbf{w}$?
* Plus, why the local loss function is defined again as $f_i$ even though it was defined as $L_i$ in line 100? Is there possibly a difference?
* Lastly, I cannot understand why the local gradient $\nabla f_c(\mathbf{w}_{t})$ comes up as in eq. (9), because the author state this equation for the description of `FedAvg` (line 227), not `FedSGD`.

## Q6. Line 291-294 and Line 791, 831
---
* What is the difference between $\zeta$ and $\hat\zeta$?
* If they are different, what is the definition of $\hat\zeta$?

## Q7. Line 324-423
---
* It is quite surprising that `FedNova`, `FedProx`, `FedBN`, and `FedDyn` fail in several heterogeneous settings of simple datasets (MNIST, CIFAR10, and Fashion-MNIST) with small clients & communication rounds. Did the authors sufficiently tune the hyperparameters of each algorithm? The listed algorithms usually _do not fail_ on this scale of datasets and assumed heterogeneity (i.e., $\beta=0.1, 0.3$) with well-tuned hyperparameters.
* I think some relevant baselines sharing similar motivation are missing: e.g., APFL[7], SuPerFed[8]. These methods also proposed to use convex combination of local and global models for mitigating client drift in terms of personalization. Please also consider adding these methods as the baselines.

[7] Adaptive Personalized Federated Learning (2020)
[8] Connecting Low-Loss Subspace for Personalized Federated Learning (2022)

**Details Of Ethics Concerns:**

While the draft makes reference to the Li et al. (2022) paper (https://arxiv.org/abs/2212.02191), I have concerns that the overall structure, motivation, exposition, and detailed components (e.g. the conformal prediction part, notations in the theoretical analysis part, and the figure in the ablation studies part) of this paper unavoidably overlap with the aforementioned source. Given the rhetorical style and sentence structure, there is a possibility that this draft could have been created by inserting [1] into LLM.

[1] On the effectiveness of partial variance reduction in federated learning with heterogeneous data (2022)

---

### Official Review · Reviewer_rpba · 2024-11-04

**Soundness:** 2
**Presentation:** 3
**Contribution:** 1
**Rating:** 3
**Confidence:** 4

**Summary:**

This paper addresses the issue of convergence slowdown in heterogenous federated learning (FL). To limit the negative impact of the clients' drift, this work proposes FedPMVR (Federated Partial Momentum Variance Reduction), that aims to better align the local updates with the global trakectory by leveraging a momentum-based partial variance reduction technique. FedPMVR only acts on the last layers of the deep neural network, which are more affected by local distribution shifts.

The paper proves the convergence properties of FedPMVR in convex and non-convex settings and empirically demonstrates the method's effectiveness in non-i.i.d. settings.

**Strengths:**

- The paper addresses a relevant issue to the FL research community
- Claims supported by theoretical convergence proofs
- Toy examples facilitate the reader's understanding
- FedPMVR does not increase communication costs, which are the main bottleneck in the deployment of FL to real-world scenarios
- Limitations explicitly addressed by the authors
- FedPMVR is easy to understand and to implement
- Code attached

**Weaknesses:**

- The paper introduces limited novelty and not enough importance is given to highlighting the differences with previous works. Specifically, FedPMVR leverages concepts which are well-known and used in the FL literature, namely variance reduction (eg, SCAFFOLD, FedPVR, FedEM, FedSVRG), momentum-based techniques to align local and global trajectories, both on the server-side (eg, FedAvgM) and the client-side (eg, FedADC, FedCM) and bias of classification layers to the distribution shifts (eg, CCVR, FedPVR). The main novelty of FedMPVR is the use of a momentum-based client-side update, limited to the last layers of the neural network, which appears incremental w.r.t. to the previously mentioned works. I believe it is important to underline the difference w.r.t. such works in terms of resulting updates (please see Questions). Overall, I believe the introduced results are not surprising, giving that FedPVMR puts together well-known beneficial approaches.
- According to Tab. 1, the improvement brought by FedPMVR is very limited (around 1 point in accuracy on average). The impact of this method would probably be more visible in more challenging scenarios (eg, more complex datasets -- see following point, or less client participation) or more complex architectures (eg, ViTs).
- Experiments are limited to small-scale image classification datasets. Examples of large-scale datasets are Landmarks-users-160k for image classification and StackOverflow for next-word prediction.


References to cited works:
[1] Karimireddy, Sai Praneeth, et al. "Scaffold: Stochastic controlled averaging for federated learning." International conference on machine learning. PMLR, 2020.
[2] Li, Bo, et al. "On the effectiveness of partial variance reduction in federated learning with heterogeneous data." Proceedings of the IEEE/CVF Conference on Computer Vision and Pattern Recognition. 2023.
[3] Dieuleveut, Aymeric, et al. "Federated-EM with heterogeneity mitigation and variance reduction." Advances in Neural Information Processing Systems 34 (2021): 29553-29566.
[4] Dawei Chen, Choong Seon Hong, Yiyong Zha, Yunfei Zhang, Xin Liu, and Zhu Han. Fedsvrg based communication efficient scheme for federated learning in mec networks. IEEE Transactions on Vehicular Technology, 70(7):7300–7304, 2021.
[5] Emre Ozfatura, Kerem Ozfatura, and Deniz Gündüz. Fedadc: Accelerated federated learning with drift control. In 2021 IEEE International Symposium on Information Theory (ISIT), pp. 467–472. IEEE, 2021. doi: 10.1109/ISIT45174.2021.9517850.
[6] Hsu, Tzu-Ming Harry, Hang Qi, and Matthew Brown. "Measuring the effects of non-identical data distribution for federated visual classification." arXiv preprint arXiv:1909.06335 (2019).
[7] Jing Xu, Sen Wang, Liwei Wang, and Andrew Chi-Chih Yao. Fedcm: Federated learning with client-level momentum, 2021.

**Questions:**

- How does the local update obtained with FedPMVR differ from the one obtained with FedPVR, SCAFFOLD and FedCM? How different is the local information contained in the updated?
- How does the client drift measure (eq. 10) change when using FedPMVR w.r.t. other approaches?
- How does FedPMVR impact the optimization path w.r.t. FedAvg, SCAFFOLD and FedPVR?
- From Fig. 2, it looks like all methods are subject to overfitting. This means the model should probably be trained for less rounds. How was the choice of rounds made?

---

### Note · Authors · 2024-11-15

I have read and agree with the venue's withdrawal policy on behalf of myself and my co-authors.